# Stigmatizing attitudes toward mental illness among mental health professionals in Kazakhstan: A cross-sectional study

Alina Abdusalyamova[1], Yerbol Nurkatov[2], Nabiy Yessimov[3] and Raushan Alibekova[4] 

[1]Master of Public Health Program, Nazarbayev University School of Medicine, Kazakhstan; [2]Department of Medicine, Nazarbayev University School of Medicine, Kazakhstan; [3]Republican Scientific and Practical Center of Mental Health, Kazakhstan and [4]Department of Biomedical Sciences, Nazarbayev University School of Medicine, Kazakhstan

## Research Article

**Keywords:**
mental health; stigma; attitudes; mental healthcare professionals; Kazakhstan

**Corresponding author:**
Raushan Alibekova;
Email: raushan.alibekova@nu.edu.kz

## Abstract

This study aimed to examine factors associated with stigmatizing attitudes toward people with mental illness among mental health professionals in Kazakhstan, the largest upper-middle-income country in Central Asia. The cross-sectional online survey was conducted among psychiatrists, narcologists (drug addiction specialists), psychotherapists, psychiatry residents and mental health nurses from all regions of Kazakhstan using the 15-item Opening Minds Stigma Scale for Healthcare Providers (OMS-HC). The mean stigma score was 40.1 of a possible 75, with higher stigma observed among older professionals ($\beta = 3.32$; 95% CI: 1.24, 5.39) and those working in the drug addiction field ($\beta = 1.99$; 95% CI: 0.32, 3.66). Male professionals had lower stigma scores ($\beta = -3.29$; 95% CI: $-5.57$, $-1.02$). Perception of mental health work as less respectable ($\beta = 6.43$; 95% CI: 3.26, 9.60) and skepticism about the positive impact of the field ($\beta = 2.69$, 95% CI: 0.52, 4.86) were associated with higher stigma; while having relatives or friends suffering from mental health issues ($\beta = -2.36$; 95% CI: $-3.95$, $-0.77$) and prior psychotherapy exposure ($\beta = -2.12$; 95% CI: $-3.98$, $-0.26$) were associated with lower levels of stigma. These findings can inform future targeted interventions on reducing stigma within healthcare settings, and promoting a more supportive and inclusive environment for both patients and providers.

## түйіндеме

Б л зерттеуді ма саты Орталы Азияда ы е ірі табысы ортадан жо ары ел - аза станда ы психикалы денсаулы са тау мамандарыны психикалы ауруы бар адамдар а деген стигматизациялаушы к з арастарасымен байланысты факторларды ба алау болып табылады. 15 тарма тан т ратын Денсаулы Са тау мамандарына арнал ан "Денсаулы Са тау Мамандары үшін Ой Ашу Стигма Шкаласын" олдана отырып, аза станны барлы айма тарынан келген психиатрлар, нарологтар, психотерапевтер, психиатриялы резиденттер ж не психикалы денсаулы са тау мейірбикелері арасында к лдене онлайн сауалнама жүргізілді. Стигманы орташа пайы 75-тен 40.1 болды, оны ішінде егде жаста ы мамандар ($\beta = 3.32$; 95% CI: 1.24, 5.39) ж не есірткіге т уелділік саласында ж мыс істейтін мамандарды арасында ($\beta = 1.99$; 95% CI: 0.32, 3.66) стигманы жо ары де гейі бай алды. Ер мамандарды ($\beta = -3.29$; 95% CI: $-5.57$, $-1.02$) стигма пайлары т мен болды. Психикалы денсаулы са тау ж мысын онша рметтелмейтін маманды ретінде абылдау ($\beta = 6.43$; 95% CI: 3.26, 9.60) ж не осы саланы о серіне күм ндану ($\beta = 2.69$, 95% CI: 0.52, 4.86) стигманы жо ары де гейімен байланысты болды; ал психикалы б зылулары бар туыстарыны немесе достарыны болуы ($\beta = -2.36$; 95% CI: $-3.95$, $-0.77$) ж не психотерапиямен б рын ы т жірибесі болуы ($\beta = -2.12$; 95% CI: $-3.98$, $-0.26$) стигманы т мен де гейімен байланысты болды. Б л т жырымдар денсаулы са тау мекемелеріндегі стигманы азайту ж не пациенттер мен медицина ызметкерлері үшін олдаушы ж не инклюзивті орта жасау үшін болаша та ма сатты араласулар а негіз бола алады. Түйін с здер: психикалы денсаулы, стигма, к з арас, психикалы денсаулы са тау мамандары, аза стан.

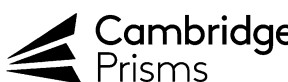

## Impact statement

Mental health stigma is characterized by prejudice, negative stereotypes and discriminatory behavior toward people with mental illness, and it remains highly prevalent and widespread across different cultures and communities. This study aimed to examine factors associated with stigmatizing attitudes of mental health professionals in Kazakhstan toward people with mental illness. Kazakhstan is the largest upper-middle-income country in the Central Asian region,

which has long been ranked among the highest for suicide rates globally. Despite the widespread prevalence of mental health stigma throughout Central Asia, there is a lack of studies on this important topic, and no prior studies have examined mental health stigma among healthcare specialists. Our study has the potential to make a significant impact by shedding light on the Kazakhstani mental health professionals' outlook toward mental illness, and informing policymakers, educators and healthcare leaders on where to focus training and awareness efforts in order to improve quality and accessibility of care, and treatment outcomes. A cross-sectional online survey was conducted among psychiatrists, narcologists, mental health nurses and psychiatry residents from all regions of Kazakhstan. The study findings highlight the persistence of stigma among mental health professionals in Kazakhstan, with overall stigma levels higher than those reported in previous studies. Key findings indicate that older professionals, those working in the drug addiction field and individuals perceiving the profession as less respectable and impactful, possess higher stigma. Gender differences were also observed, with male professionals reporting lower stigma. Those with personal experiences of mental illness and engagement in psychotherapy demonstrated lower stigma levels. Study findings emphasize the critical need for directed efforts to combat stigma, particularly for professionals working with addictions. Addressing occupational stigma, promoting psychotherapy training and strengthening the professional perception of mental health workers are essential steps toward improving attitudes toward patients with mental illness and patient care.

## Introduction

Mental health stigma is a complex and pervasive issue that negatively impacts individuals, communities and healthcare systems. It has a deep-rooted past and is widespread in numerous societies (Rössler, 2016).

The stigma surrounding mental health conditions plays a significant role in marginalizing and discriminating against those affected, hindering their access to critical resources and limiting their ability to engage fully in society. It also disrupts nearly all areas of mental healthcare, from health promotion and early support to the availability and consistency of treatment (Sirey et al., 2001; Thornicroft, 2006, 2008; Mehta et al., 2015). Stigma toward mental illness can damage self-esteem, diminish quality of life, reduce social support and lower treatment adherence, thereby worsening overall mental health (Livingston and Boyd, 2010; Fox et al., 2018). Furthermore, stigma intensifies suicidal thoughts and behaviors, especially in those already at risk, and correlates with higher suicide and mortality rates (Thornicroft, 2006; Taylor et al., 2011; Rimkeviciene et al., 2015; Schomerus et al., 2015).

There is a growing interest in research on stigma and social exclusion related to mental health among different populations (Evans-Lacko et al., 2014). Stigmatizing attitudes vary by cultural and social context. Stigma toward people with mental illness remains prevalent in the Nordic region, despite strong welfare systems and equal access to healthcare (Högberg et al., 2012; Hellström et al., 2022). People with schizophrenia and psychotic disorders experienced the highest levels of stigma, more than those with depression or autism (Jensen et al., 2016; Svensson and Hansson, 2016). While in the United States, stigma around depressive disorders resembles the situation in Nordic countries and has significantly decreased, stigma reduction efforts have had limited success in changing behaviors, especially for schizophrenia and alcohol dependence (Pescosolido et al., 2021). In Asian nations and low- and middle-income countries, stigma toward mental illness is highly prevalent. In contrast with the European region, cultural beliefs and myths play a significant role; for example, mental illness is linked to karma, supernatural forces or moral weakness. In addition, older age, male gender and lower socioeconomic status were linked to higher levels of stigma (Vaishnav et al., 2023). There is limited research on the topic in post-Soviet Union countries and the Central Asian region (Aliev et al., 2021; Quirke et al., 2021).

Research on stigma among mental health professionals presents mixed findings. Some studies suggest that mental healthcare providers hold more positive attitudes than the general population (Lauber et al., 2004; Hori et al., 2011; Hellström et al., 2022;

Vaishnav et al., 2023). Conversely, other studies indicate that their biases are comparable to those found in the general public (Schulze, 2007; Jensen et al., 2016). Additionally, some research suggests that mental health professionals may, in certain cases, exhibit even higher levels of stigma. For instance, psychiatrists have been reported to endorse more negative stereotypes about individuals with mental illness than both the general population and other mental health professionals (Lauber et al., 2000; Nordt et al., 2006; Henderson et al., 2014). Notably, there is evidence that psychiatrists themselves may reinforce stigmatizing beliefs – for example, associating individuals with schizophrenia with traits such as "unpredictability" and "dangerousness" (Lauber et al., 2004). This highlights a paradox: mental health professionals serve as both advocates for change and, at times, contributors to stigma (Schulze, 2007).

Research on the stigmatizing attitudes of mental health professionals remains scarce, with even fewer studies exploring the underlying factors influencing these perceptions. Burnout and lower job satisfaction have been pointed out as contributing factors to stigma among mental health professionals (Gibb et al., 2010; Solmi et al., 2020; Yavuz et al., 2020; Vaishnav et al., 2023). Findings on age and stigma among psychiatrists are conflicting. Some studies link younger age to greater negative stereotyping, especially toward patients with schizophrenia (Loch et al., 2013b), while others associate older age, higher education and longer patient contact with more favorable attitudes (Vibha et al., 2008). However, longer careers in psychiatry have also been linked to higher stigma levels (Nordt et al., 2006; da Silva et al., 2021), with older psychiatrists more likely to hold stigmatizing attitudes (Őri et al., 2023a).

Attitudes toward mental illness vary across healthcare professionals and levels of training. Among trainees, medical students often hold opposing views, underscoring gaps in education (Vaishnav et al., 2023). However, those considering a career in psychiatry, along with psychiatrists themselves, tend to display more favorable attitudes (Reavley et al., 2013; Janoušková et al., 2021; Őri et al., 2022). Although research on nurses' attitudes remains limited, some findings suggest that community and public health nurses demonstrate more accepting views, whereas inpatient nurses are more socially restrictive toward individuals with mental illness (Ishige and Hayashi, 2005; Linden and Kavanagh, 2012).

Mental health professionals' attitudes toward mental illness can be influenced by personal experiences and exposure. Those who are hesitant to seek mental healthcare for themselves tend to exhibit stronger stigmatizing views (Corrigan et al., 2014). Conversely, psychiatrists with a history of psychiatric diagnosis or treatment generally demonstrate more accepting attitudes (Yavuz et al., 2020; Őri et al., 2022, 2023a). Regular interactions with individuals with

mental illness in community settings have also been associated with reduced stigma among psychiatrists (Yavuz et al., 2020; da Silva et al., 2021). However, in some cases, frequent exposure to mental illness has been linked to heightened stigmatizing views within the mental health professional community (Nordt et al., 2006; da Silva et al., 2021).

Beyond personal experience, a professional environment also plays a role in shaping attitudes. Psychiatrists who provided psychotherapy and worked in settings with less stigmatizing colleagues were significantly more likely to hold positive views toward individuals with mental illness (Őri et al., 2022, 2023a). However, levels of stigma may also be influenced by broader socio-cultural factors. Studies suggest that psychiatrists in post-communist countries exhibit higher levels of stigma compared to their counterparts elsewhere (Harangozo et al., 2014), particularly high stigma scores were reported among psychiatrists in Latvia, Ukraine and Belarus (Őri et al., 2023a).

There is a lack of comprehensive data on how stigma among mental health professionals differs across cultural contexts, particularly in non-Western countries, emphasizing the need for more globally representative research.

Stigma is reported to be widespread throughout Central Asia, and it is characterized by prevalent societal misunderstandings about mental health (Aliev et al., 2021). Kazakhstan is the largest upper-middle-income country in the Central Asian region, which has been ranked among the highest for suicide rates globally, particularly affecting adolescents (WHO, 2025). The country implemented its first national suicide prevention strategy in 2014, drawing attention to a deep-rooted reluctance among families to assist individuals dealing with mental health issues, including suicidal tendencies, due to prevailing fears of social judgment and stigma. Destigmatization became a central focus of the prevention effort in the country, supported by technical assistance from UNICEF (WHO, 2022) however, no prior studies have examined mental health stigma among healthcare specialists.

The research aims of this study were to (1) investigate how prevalent stigma toward mental illness is among mental health specialists (psychiatrists, narcologists, psychotherapists, psychiatry residents and mental health nurses) in Kazakhstan, and (2) what factors contribute to the stigmatizing attitudes. Understanding the stigma toward mental illness among healthcare professionals is essential, as it directly influences the quality of care patients receive. Study findings can provide insights into the barriers to effective mental healthcare and inform interventions aimed at improving attitudes toward patients with mental illness among medical workers.

## Methods

### Study design and sample

This study used a cross-sectional design, collecting data through a self-administered online questionnaire shared via secure links. The gathered data were analyzed quantitatively to spot key patterns in attitudes of mental health professionals toward mental illness. The reporting of this observational study follows the STROBE guidelines.

The questionnaire was distributed among psychiatrists, narcologists, psychotherapists, psychiatry residents and mental health nurses working in various healthcare institutions across different regions of Kazakhstan. The recruitment message, including a link to the online survey, was distributed among mental health specialists using purposive, snowball and convenience sampling methods. Social media platforms such as WhatsApp, Instagram and

Telegram were utilized to distribute the link. Specifically, the survey link was shared in group chats on WhatsApp for mental health professionals from key hospitals and clinics. The survey was sent to direct contacts of mental health specialists, who were encouraged to share the link with colleagues to maximize the response rate.

With a total population of 4,561 mental health professionals in Kazakhstan (803 psychiatrists, 3,312 nurses, 262 psychologists and 184 child psychiatrists) (WHO, 2022), the sample size for the study was calculated as 302 participants, using the formula: $n = [DEFF*Np(1-p)]/[(d^2/Z^2_{1-\alpha/2}*(N-1) + p*(1-p)]$. The total population size, used to apply the finite population correction factor (FPC), was 4,561. The expected prevalence of the outcome in the population was estimated at 30%, with a margin of error of ±5%. The confidence level was set at 95%, corresponding to an absolute precision of 5%. A design effect (DEFF) of 1 was assumed.

Inclusion criteria include currently practicing mental health professionals with direct patient care experience.

### Procedure

The survey was developed based on the literature review of previous studies (Kassam et al., 2012; Modgill et al., 2014; Őri et al., 2022, 2023a). The questionnaire was built into the Qualtrics survey platform and was developed in English, Kazakh and Russian. The link was shared with mental health specialists through purposive and snowball sampling. The online survey was conducted between December 2024 and February 2025.

The questionnaire comprised five modules with 33 questions in total. The modules were (1) sociodemographic characteristics, (2) professional characteristics and beliefs, (3) stigmatizing attitudes toward people with mental illness, (4) lived experience of mental illness and (5) job satisfaction and burnout.

### Measures

#### Sociodemographic characteristics
This section included four questions inquiring about participants' age, gender, ethnicity and region of residence.

#### Professional characteristics and beliefs
Questions included years of experience in mental health, year of graduation, qualification status (trainee or specialist), a field of psychiatry (child, adult, narcology, psychotherapy, mental health nursing), type of healthcare setting (inpatient, outpatient, private practice), perceived respectability of mental health work and perceived stigma toward people with mental illness among colleagues.

#### Stigmatizing attitudes toward mental illness
To evaluate the main outcome of the study, the 15-item Opening Minds Stigma Scale for Healthcare Providers (OMS-HC) was employed. Initially, this scale was developed as a 20-item scale; however, the 15-item version of the OMS-HC is considered superior because a previous validation study using factor analysis identified five items that could be dropped without compromising the scale's effectiveness (Modgill et al., 2014). Each item was scored on a 5-point Likert scale, where 1 represents "strongly disagree" and 5 means "strongly agree." Questions 2, 6, 7, 8 and 14 were reverse-scored as per the OMS-HC guidelines. The overall score indicates participants' general stigma levels, while its three subscales measure different aspects, including Attitudes, Disclosure and Help-Seeking and Social Distance subscales. Stigma was quantified by adding the responses to all 15 items on the scale, yielding a total score that could

range between 15 and 75. Higher scores signify greater stigma. Prior research has indicated that the scale demonstrated a satisfactory level of internal consistency, with Cronbach's alpha = 0.79, with subscales scoring 0.67–0.68 (Modgill et al., 2014). The subscales' lower alpha may result from fewer items, which can reduce reliability below the recommended 0.70 threshold (Streiner, 2003).

The validated Russian version of the scale was requested via e-mail from Dr. Dorottya Őri, a principal investigator of a similar study across 32 European countries (Őri et al., 2022, 2023a). The psychometric properties of the OMS-HC were examined across all participating countries using a series of confirmatory factor analyses (Őri et al., 2023b), which indicated that the total score, rather than subscale scores, is recommended for assessing overall stigmatizing attitudes. The OMS-HC was translated into the Kazakh language employing the forward- and back-translation method by our bilingual research team members (Appendix 1, Supplementary Materials). We thoroughly adopted the scale with permission from developers, and tested for its reliability, face and content validity in local languages. Internal consistency of the OMS-HC scale within our study sample was evaluated using Cronbach's alpha coefficient, which was 0.66 for the Russian version and 0.65 for the Kazakh version. The OMS-HC scale and questionnaire were pretested with a small group of mental health professionals, resulting in minor wording adjustments for clarity.

### Lived experience of mental illness

This section was comprised of questions on the presence of friends or family members with mental illness, help-seeking behavior for one's own mental health conditions, participation in psychotherapy for any reason and personal exposure to medical treatment for mental illness.

### Job satisfaction and burnout

Job satisfaction was assessed using a single-item measure on a continuous scale ranging from 0 to 10, in response to the question: *"On a scale from 0 to 10, how would you rate the level of your job satisfaction?"* A score of 0 indicated "Very Dissatisfied," while a score of 10 indicated "Very Satisfied." To measure burnout, the non-proprietary single-item burnout subscale was used, which allows respondents to interpret burnout in their own terms, asking: "Overall, based on your definition of burnout, how would you rate your level of burnout?" Answers follow a five-point ordinal scale, ranging from 1 ("I enjoy my work and experience no burnout symptoms") to 5 ("I feel completely burned out and may need to make changes or seek help"). Intermediate options describe increasing levels of stress and exhaustion, from occasional fatigue (2) to persistent symptoms and workplace frustration (4) (Schmoldt et al., 1994). For analysis, responses were categorized into two groups: scores ≤2 indicate no burnout, while scores ≥3 suggest one or more symptoms of burnout. Several studies assessed its validity by comparing it to the full Maslach Burnout Inventory- Human Services Survey (MBI-HSS), concluding that it effectively identified burnout, especially in relation to emotional exhaustion (MBI-EE), while offering a more efficient and license-free alternative for data collection (Rohland et al., 2004; Hansen and Girgis, 2010; Dolan et al., 2015).

### Statistical analysis

Data cleaning and analyses were performed using Stata software, version 18.0. Responses with missing data on the OMS-HC scale, as well as those containing only sociodemographic information, were excluded from the analysis; however, for other variables (sociodemographic characteristics, professional characteristics and beliefs, lived experience of mental illness, job satisfaction and burnout), cases with partial missing data were retained to avoid unnecessary reduction in sample size. Univariate analysis was conducted to examine the variables and describe participant characteristics, presenting categorical variables as frequencies and percentages, while continuous variables were summarized as mean values with standard deviation.

Bivariate analysis was then performed to explore associations between the main outcome variable, the attitudes toward mental illness assessed using the OMS-HC 15-item scale, and independent variables, including sociodemographic factors, professional data, personal experience of mental illness, job satisfaction and burnout and attitudes toward mental health recovery, using a significance threshold of $p < 0.05$. Simple linear regression analyses were applied to assess relationships between the OMS-HC score and independent variables. Due to low response counts in certain subcategories of independent variables and recommended guidelines, some of them were merged into broader categories.

Subsequently, multiple linear regression analysis was conducted, and after adjusting for covariates, only independent variables maintaining significance at $p < 0.05$ remained in the final model. To address multicollinearity, the variance inflation factor (VIF) was used, and variables were assessed for interactions and confounding to ensure the robustness of the model.

## Results

### Descriptive analysis

Table 1 summarizes the sociodemographic and professional profiles of the study participants, encompassing age, gender, ethnicity, region of residence, years of work experience, graduation year, qualifications, field of work, job setting, job satisfaction and burnout levels.

Overall, 388 responses were recorded, and after data cleaning, 348 of them were included in the analysis. The majority of responses were in the Russian language ($n = 326$), followed by responses in Kazakh ($n = 20$) and English ($n = 2$). Participants' ages ranged from 18 to over 65 years, with a minimum age of 22 and a maximum of 74 years old. The majority of participants were aged 45–54 years (22.99%), followed by those aged 55–64 years (19.54%). Regarding gender distribution, the sample consisted of 85.59% female ($n = 297$) and 14.41% male ($n = 50$). Participants were mainly Kazakh (59.42%), followed by Russian (32.46%) and other ethnicities (8.12%). Participants were from various regions across Kazakhstan, with the highest proportion from Kostanay Region (50.58%), followed by Almaty City (11.63%) and East Kazakhstan Region (10.47%). Other regions were represented in smaller proportions.

Most participants had 11–20 years of work experience (31.30%), followed by those with 1–5 years (22.17%) and 21–30 years (20.43%). Regarding graduation year, nearly half of the participants (49.56%) graduated after the year 2000, while 31.27% graduated before 1991. Predominantly, participants held the qualification of Specialist (93.27%), with a small proportion being trainees (6.73%). The highest percentage of participants worked in adult psychiatry (39.13%), followed by narcology (33.62%) and mental health nursing (16.52%). Fewer participants worked in child psychiatry (2.90%), psychotherapy (3.48%) or both in psychiatry and narcology (1.74%).

Work settings varied, with 62.79% of participants working in inpatient hospitals, followed by 16.86% in psychiatric outpatient services. Smaller proportions worked in psychosocial rehabilitation departments (4.94%), multiple settings (4.36%) or exclusively private practice (2.03%). The mean job satisfaction score was

**Table 1.** Sociodemographic and professional characteristics of participants (*n* = 348)

| Variables | Frequency[§]/ Mean | Percent/ SD |
|---|---|---|
| Language | | |
| English | 2 | 0.57 |
| Kazakh | 20 | 5.75 |
| Russian | 326 | 93.68 |
| Age | | |
| 18–24 | 7 | 2.8 |
| 25–34 | 51 | 20.4 |
| 35–44 | 39 | 15.6 |
| 45–54 | 80 | 32 |
| 55–64 | 68 | 27.2 |
| >65 | 5 | 2 |
| Gender | | |
| Female | 297 | 85.59 |
| Male | 50 | 14.41 |
| Ethnicity | | |
| Kazakh | 205 | 59.42 |
| Russian | 112 | 32.46 |
| Other | 28 | 8.12 |
| Region | | |
| Kostanay Region | 174 | 50.58 |
| Almaty City | 40 | 11.63 |
| East Kazakhstan Region | 36 | 10.47 |
| Astana City | 25 | 7.27 |
| Atyrau Region | 25 | 7.27 |
| Almaty Region | 16 | 4.65 |
| Zhetysu Region | 8 | 2.33 |
| Shymkent City | 7 | 2.03 |
| Kyzylorda Region | 5 | 1.45 |
| Abai Region | 3 | 0.87 |
| Turkestan Region | 2 | 0.58 |
| Aktobe Region | 1 | 0.29 |
| Karaganda Region | 1 | 0.29 |
| North Kazakhstan Region | 1 | 0.29 |
| Work experience (years) | | |
| 1–5 | 51 | 22.17 |
| 6–10 | 34 | 14.78 |
| 11–20 | 72 | 31.30 |
| 21–30 | 47 | 20.43 |
| ≥31 | 26 | 11.30 |
| Graduation year | | |
| Before 1991 | 106 | 31.27 |
| Between 1991–2000 | 65 | 19.17 |

(*Continued*)

**Table 1.** (*Continued*)

| Variables | Frequency[§]/ Mean | Percent/ SD |
|---|---|---|
| After 2000 | 168 | 49.56 |
| Qualification | | |
| Specialist | 305 | 93.27 |
| Trainee | 22 | 6.73 |
| Field of work | | |
| Adult psychiatry | 135 | 39.13 |
| Child psychiatry | 10 | 2.90 |
| Psychotherapy | 12 | 3.48 |
| Mental health nursing | 57 | 16.52 |
| Narcology | 116 | 33.62 |
| Psychiatry and narcology | 6 | 1.74 |
| Other | 9 | 2.61 |
| Setting | | |
| Inpatient hospital | 216 | 62.79 |
| Psychiatric outpatient service | 58 | 16.86 |
| Psychosocial rehabilitation department | 17 | 4.94 |
| Multiple places | 15 | 4.36 |
| Inpatient hospital, Psychiatric outpatient service | 13 | 3.78 |
| I do not work with patients | 11 | 3.20 |
| Exclusively private practice | 7 | 2.03 |
| Other outpatient services where psychiatric patients are also treated | 7 | 2.03 |
| Job satisfaction | | |
| Min–0, max–10 | 7.42 | 2.24 |
| Burnout | | |
| No symptoms of burnout | 254 | 74.27 |
| 1 or > symptoms of burnout | 88 | 25.73 |

[§]Total frequencies may vary because of the missing values.

7.42 (SD = 2.24) on a 10-point scale, indicating overall moderate to high job satisfaction among participants. Regarding burnout symptoms, 74.27% of participants reported no symptoms of burnout, while 25.73% experienced at least one symptom of burnout.

Table 2 presents findings on participants' professional values and beliefs about mental healthcare, and personal experiences with mental illness. Participants expressed diverse opinions regarding the respectability of working in the mental health field compared to other health and social care professions. While 19.65% strongly disagreed and 32.66% disagreed with the notion that mental health work is less respectable, a considerable proportion remained neutral (17.05%). However, 24.28% agreed and 6.36% strongly agreed with this perception.

Regarding stigmatizing attitudes among close colleagues toward patients with mental illness, more than half of the participants (55.26%) reported perceiving no stigmatization at all, while 17.25% acknowledged a small extent of stigmatization. Notably, 18.42% believed stigma existed to some extent and 9.06% perceived stigma to a great extent among their colleagues.

**Table 2.** Professional values and beliefs, and personal experience of mental illness

| Variables | Frequency | Percent |
|---|---|---|
| *Professional values and beliefs* | | |
| Believe that working in the mental health field is less respectable than working in other fields of health and social care. | | |
| Strongly disagree | 68 | 19.65 |
| Disagree | 113 | 32.66 |
| Neither agree nor disagree | 59 | 17.05 |
| Agree | 84 | 24.28 |
| Strongly agree | 22 | 6.36 |
| Do you believe there is a stigmatizing attitude among your close colleagues toward patients with mental illness? | | |
| Not at all | 189 | 55.26 |
| To small extent | 59 | 17.25 |
| To some extent | 63 | 18.42 |
| To great extent | 31 | 9.06 |
| Do you believe that mental healthcare contributes to the health of people, families, communities and society in unique and meaningful ways? | | |
| Strongly disagree | 18 | 5.29 |
| Disagree | 24 | 7.06 |
| Neither agree nor disagree | 54 | 15.88 |
| Agree | 172 | 50.59 |
| Strongly agree | 72 | 21.18 |
| *Personal experience of mental illness* | | |
| Do you have friends or family members with a mental illness? | | |
| No | 222 | 64.53 |
| I do not know | 19 | 5.52 |
| Prefer not to answer | 11 | 3.20 |
| Yes | 92 | 26.74 |
| Have you ever sought help for your own mental health conditions? | | |
| No | 297 | 85.84 |
| Prefer not to answer | 7 | 2.02 |
| Yes | 42 | 12.14 |
| Have you ever been participating in psychotherapy for any reason? | | |
| No | 270 | 78.03 |
| Prefer not to answer | 7 | 2.02 |
| Yes | 69 | 19.94 |
| Have you ever been medically treated for a mental illness? | | |
| No | 319 | 93.00 |
| Prefer not to answer | 5 | 1.46 |
| Yes | 19 | 5.54 |

When asked whether mental healthcare contributes uniquely and meaningfully to society, a majority of participants held positive views, with 50.59% agreeing and 21.18% strongly agreeing. However, 5.29% strongly disagreed and 7.06% disagreed, while 15.88% remained neutral on this statement.

A significant proportion of participants (64.53%) reported not having friends or family members with mental illness, while 26.74% confirmed having such connections. A small proportion of respondents were uncertain (5.52%) or preferred not to answer (3.20%). Regarding seeking help for personal mental health conditions, 85.84% stated they had never sought professional help, while 12.14% reported having done so and 2.02% preferred not to disclose this information.

Participation in psychotherapy for any reason was reported by 19.94% of respondents, whereas 78.03% had never participated in psychotherapy and 2.02% preferred not to answer.

When asked about medical treatment for a mental illness, 93.00% reported never receiving such treatment, while 5.54% had been medically treated and 1.46% preferred not to disclose this information.

Table 3 presents the descriptive statistics for the 15-item OMS-HC. The data were normally distributed, with a total mean of 40.1 (SD = 6.23; range: 18–57).

For the Attitudes of Healthcare Providers Toward People with Mental Illness subscale, the mean item scores ranged from 1.94 to 3.27, with a subtotal mean of 14.02 (SD = 3.62; range: 6–30). The highest-rated item, "More than half of people with mental illness don't try hard enough to get better," had 3.27 (SD = 1.08), while the lowest-rated item, "Despite my professional beliefs, I have negative reactions toward people who have mental illness," had 1.94 (SD = 0.79).

In the Disclosure and Help-Seeking subscale, a subtotal mean was 11.25 (SD = 2.42; range: 4–18). The majority of respondents were reluctant to disclose their own mental health concerns, resulting in a right-skewed distribution of responses. This trend was particularly evident in items such as "If I had a mental illness, I would tell my friends" (mean = 3.17), "If I were under treatment for a mental illness, I would not disclose this to any of my colleagues" (mean = 2.98) and "I would see myself as weak if I had a mental illness and could not fix it myself" (mean = 2.85).

The Social Distance subscale had a subtotal mean of 14.9 (SD = 3.06; range: 5–23), with item scores generally tending toward higher values. Notably, respondents expressed the greatest concern regarding individuals with mental illness working with children (mean = 3.50), followed by willingness to visit a physician who had been treated for a mental illness (mean = 3.11) and acceptance of having a neighbor with a mental illness (mean = 3.09). These findings indicate that while some respondents were open to social integration, reservations remained, particularly in professional and caregiving contexts.

### Bivariate analysis

A simple linear regression analysis was conducted to examine the associations between various independent factors and stigma levels, as measured by the OMS-HC scale (Table 4).

Age was found to be positively associated with stigma levels, with older age groups demonstrating significantly higher stigma scores compared to the reference group (18–34 years). Participants aged 35–44 years ($\beta$ = 3.10, $p$ = 0.023), 45–54 years ($\beta$ = 3.09, $p$ = 0.004) and ≥ 55 years ($\beta$ = 4.18, $p$ < 0.001) had significantly higher stigma levels.

Male participants exhibited significantly lower stigma scores than females ($\beta$ = −3.53, $p$ < 0.001). Additional analyses of OMS-HC subscales revealed statistically significant gender differences on Attitudes and Social Distance subscales, with males reporting lower stigma scores ($\beta$ = −1.35, $p$ = 0.015 and $\beta$ = −1.67, $p$ < 0.001,

**Table 3.** Opening minds stigma scale for healthcare providers

| Question | Mean | SD |
|---|---|---|
| *Attitudes of healthcare providers toward people with mental illness* | | |
| I am more comfortable helping a person who has a physical illness than I am helping a person who has a mental illness. | 2.24 | 1.03 |
| Despite my professional beliefs, I have negative reactions toward people who have mental illness. | 1.94 | 0.79 |
| There is little I can do to help people with mental illness. | 2.29 | 1.00 |
| More than half of people with mental illness do not try hard enough to get better. | 3.27 | 1.08 |
| Healthcare providers do not need to be advocates for people with mental illness. | 2.16 | 0.9 |
| I struggle to feel compassion for a person with a mental illness. | 2.18 | 0.98 |
| Subscale score | 14.02 (min–6, max–30) | 3.62 |
| *Disclosure/help-seeking* | | |
| If I were under treatment for a mental illness I would not disclose this to any of my colleagues. | 2.98 | 1.00 |
| I would see myself as weak if I had a mental illness and could not fix it myself. | 2.85 | 1.06 |
| I would be reluctant to seek help if I had a mental illness. | 2.25 | 0.99 |
| If I had a mental illness, I would tell my friends. | 3.17 | 1.00 |
| Subscale score | 11.25 (min–4, max–18) | 2.42 |
| *Social distance* | | |
| If a colleague with whom I work told me they had a managed mental illness, I would be as willing to work with him/her. | 2.37 | 1.01 |
| Employers should hire a person with a managed mental illness if he/she is the best person for the job. | 2.85 | 1.06 |
| I would still go to a physician if I knew that the physician had been treated for a mental illness. | 3.11 | 1.01 |
| I would not want a person with a mental illness, even if it were appropriately managed, to work with children. | 3.5 | 1.07 |
| I would not mind if a person with a mental illness lived next door to me. | 3.09 | 0.98 |
| Subscale score | 14.9 (min–5, max–23) | 3.06 |
| Total score | 40.1 (min–18, max–57) | 6.23 |

respectively; not shown in the table), while there was no significant difference on the Disclosure/Help-seeking subscale.

Regional differences in stigma levels were observed, with mental health professionals from Kostanay Region reporting significantly higher stigma compared to those in Astana City ($\beta = 2.80$, $p = 0.03$). In contrast, professionals from Zhetysu Region demonstrated significantly lower stigma ($\beta = -4.78$, $p = 0.05$), with the largest reduction observed among all regions, although marginally

significant. No significant differences were found for other regions when compared to Astana City.

Mental health professionals working in mental health nursing ($\beta = 2.67$, $p = 0.01$) and narcology ($\beta = 3.09$, $p < 0.001$) had significantly higher stigma levels compared to those in psychiatry/psychotherapy. Graduation year was also associated with stigma, as those who graduated after 2000 had significantly lower stigma scores ($\beta = -2.65$, $p = 0.001$) in comparison with specialists who graduated before 1991. However, ethnicity, work experience and job satisfaction did not show significant associations with stigma levels.

Participants who had friends or family members with mental illness exhibited significantly lower stigma levels ($\beta = -2.65$, $p = 0.001$) than those who did not. Similarly, individuals who had sought professional help for their own mental health ($\beta = -2.09$, $p = 0.045$) or had participated in psychotherapy ($\beta = -3.65$, $p < 0.001$) reported significantly lower stigma scores.

Regarding professional beliefs, there was a meaningful difference across participants who strongly agreed with the idea that mental healthcare contributes meaningfully to society and those who just agreed ($\beta = 2.31$, $p = 0.008$) or disagreed ($\beta = 4.38$, $p < 0.001$).

Furthermore, individuals who expressed neutrality ($\beta = 4.48$, $p < 0.001$), agreement ($\beta = 2.88$, $p = 0.004$) or strong agreement ($\beta = 4.76$, $p = 0.001$) with the perception that working in mental health is less respectable exhibited significantly higher stigma levels.

## Multivariate analysis

The final multivariate linear regression model is expressed as:

*Stigma level (OMS-HC) = $\beta_0$ + $\beta_1$Gender + $\beta_2$Age + $\beta_3$Field of work + $\beta_4$Proffesional stigma + $\beta_5$Mental illness in friends&family + $\beta_6$Exposure to psychotherapy + $\beta_7$Contribution of the field.*

The model explained 33.5% of the variance in stigma levels among mental health professionals ($R^2 = 0.335$, adjusted $R^2 = 0.2875$, $F(15, 210) = 7.05$, $p < 0.001$). No significant interactions were observed among the study variables.

Age remained a significant predictor, with professionals aged ≥55 exhibiting significantly higher stigma levels compared to the youngest age group ($\beta = 3.32$, $p = 0.002$) (Table 5). Gender differences persisted, with male professionals demonstrating significantly lower stigma scores than their female counterparts ($\beta = -3.29$, $p = 0.005$). Field of work also played a role, as professionals in narcology had significantly higher stigma scores compared to those in psychiatry/psychotherapy ($\beta = 1.99$, $p = 0.02$).

Perceived lack of respectability of career in mental health strongly predicted higher stigma, with neutrality ($\beta = 3.50$, $p = 0.005$), agreement ($\beta = 3.53$, $p = 0.002$) and strong agreement ($\beta = 6.43$, $p < 0.001$) linked to increased stigma levels. Personal experience with mental illness was associated with lower stigma, with those having affected friends or family ($\beta = -2.36$, $p = 0.004$) and psychotherapy exposure ($\beta = -2.12$, $p = 0.03$) showing reduced stigma. Conversely, weaker belief in their field's positive contribution was associated with higher stigma.

## Discussion

This study explores the often-overlooked issue of stigma within the mental health profession. Given the limited research on this topic, particularly among mental healthcare providers in Central Asia,

**Table 4.** Simple linear regression analysis of the associations of independent variables with stigma level among mental health professionals

| Variables | Coefficient (β) | p-value | 95% CI |
|---|---|---|---|
| Age | | | |
| 18–34 | Ref | | |
| 35–44 | 3.10 | 0.023* | [0.44, 5.77] |
| 45–54 | 3.09 | 0.004** | [0.97, 5.21] |
| ≥55 | 4.18 | <0.001*** | [1.96, 6.4] |
| Gender | | | |
| Female | Ref | | |
| Male | −3.53 | <0.001*** | [−5.41, −1.66] |
| Ethnicity | | | |
| Kazakh | Ref | | |
| Russian | −0.15 | 0.84 | [−1.64, 1.34] |
| Other | −1.16 | 0.38 | [−3.77, 1.46] |
| Region | | | |
| Astana City | Ref | | |
| Kostanay Region | 2.80 | 0.03* | [0.22, 5.38] |
| Almaty City | 0.34 | 0.83 | [−2.73, 3.40] |
| East Kazakhstan Region | 0.26 | 0.87 | [−2.87, 3.39] |
| Atyrau Region | 3.07 | 0.08 | [−0.39, 6.52] |
| Almaty Region | 1.97 | 0.31 | [−1.86, 5.80] |
| Zhetysu Region | −4.78 | 0.05 | [−9.64, 0.08] |
| Shymkent City | 4.72 | 0.11 | [−1.14, 10.58] |
| Kyzylorda Region | 5.22 | 0.11 | [−1.22, 11.66] |
| Abai Region | −4.61 | 0.22 | [−11.92, 2.69] |
| Turkestan Region | 7.22 | 0.107 | [−1.56, 16.00] |
| Aktobe Region | −1.28 | 0.84 | [−13.47, 10.91] |
| Karaganda Region | −3.28 | 0.60 | [−15.47, 8.91] |
| North Kazakhstan Region | −1.28 | 0.84 | [−13.47, 10.91] |
| Experience (years) | | | |
| 1–10 | Ref | | |
| 11–20 | 1.56683 | 0.14 | [−0.5, 3.63] |
| ≥ 21 | .629563 | 0.54 | [−1.41, 2.67] |
| Year of graduation | | | |
| Before 1991 | Ref | | |
| Between 1991–2000 | −1.09 | 0.28 | [−3.04, 0.87] |
| After 2000 | −2.65 | 0.001** | [−4.22, −1.07] |
| Qualification | | | |
| Specialist | Ref | | |
| Trainee | −0.51 | 0.72 | [−3.29, 2.28] |
| Field of work | | | |
| Psychiatry/Psychotherapy | Ref | | |
| Mental health nursing | 2.67 | 0.01* | [0.64, 4.69] |
| Narcology | 3.09 | <0.001*** | [1.60, 4.59] |

(*Continued*)

**Table 4.** (*Continued*)

| Variables | Coefficient (β) | p-value | 95% CI |
|---|---|---|---|
| Other | −0.55 | 0.75 | [−3.88, 2.79] |
| Setting | | | |
| Inpatient hospital | Ref | | |
| Psychiatric outpatient service | −0.11 | 0.90 | [−1.97, 1.74] |
| Psychosocial rehabilitation department | 0.79 | 0.63 | [−2.38, 3.95] |
| Several places | −1.87 | 0.28 | [−5.23, 1.50] |
| Inpatient hospital, Psychiatric outpatient service | −4.07 | 0.02* | [−7.56, −0.59] |
| I do not work with patients | 2.35 | 0.24 | [−1.60, 6.30] |
| Exclusively private practice | −0.44 | 0.86 | [−5.12, 4.25] |
| Other outpatient services where psychiatric patients are also treated | 1.71 | 0.47 | [−2.98, 6.39] |
| Job satisfaction | 0.19 | 0.25 | [−0.14, 0.52] |
| Burnout | | | |
| No symptoms of burnout | Ref | | |
| 1 or > symptoms of burnout | −0.80 | 0.32 | [−2.37, 0.77] |
| Contribution of the field | | | |
| Strongly agree | Ref | | |
| Agree | 2.31 | 0.01* | [0.60, 4.02] |
| Disagree | 4.38 | <0.001*** | [2.48, 6.29] |
| Professional stigma | | | |
| Strongly disagree | Ref | | |
| Disagree | 0.36 | 0.70 | [−1.51, 2.24] |
| Neither agree nor disagree | 4.48 | <0.001*** | [2.28, 6.69] |
| Agree | 2.88 | 0.004** | [0.90, 4.87] |
| Strongly agree | 4.76 | 0.001** | [1.85, 7.68] |
| Stigma among colleagues | | | |
| Not at all | Ref | | |
| To small extent | −0.06 | 0.95 | [−1.96, 1.83] |
| To some extent | 0.09 | 0.92 | [−1.73, 1.91] |
| To great extent | −1.69 | 0.17 | [−4.09, 0.71] |
| Mental illness in friends & family | | | |
| No | Ref | | |
| Yes | −2.65 | 0.001** | [−4.15, −1.15] |
| Sought professional help | | | |
| No | Ref | | |
| Yes | −2.09 | 0.04* | [−4.13, −0.05] |
| Exposure to psychotherapy | | | |
| No | Ref | | |
| Yes | −3.65 | <0.001*** | [−5.29, −2.02] |

*Significance level $p < 0.05$.
**Significance level $p < 0.01$.
***Significance level $p < 0.001$.

**Table 5.** Multiple linear regression analysis of the associations of independent variables with stigma level among mental health professionals

| Variables | Coefficient (B) | p-value | 95% CI |
|---|---|---|---|
| Age | | | |
| 18–34 | Ref | | |
| 35–44 | 2.06 | 0.10 | [−0.38, 4.51] |
| 45–54 | 1.55 | 0.13 | [−0.45, 3.55] |
| ≥55 | 3.32 | 0.002** | [1.24, 5.39] |
| Gender | | | |
| Female | Ref | | |
| Male | −3.29 | 0.01* | [−5.57, −1.02] |
| Field of work | | | |
| Psychiatry/ Psychotherapy | Ref | | |
| Mental health nursing | 0.52 | 0.66 | [−1.82, 2.86] |
| Narcology | 1.99 | 0.02* | [0.32, 3.66] |
| Other | −1.37 | 0.45 | [−4.93, 2.20] |
| Professional stigma | | | |
| Strongly disagree | Ref | | |
| Disagree | 0.28 | 0.79 | [−1.78, 2.34] |
| Neither agree nor disagree | 3.50 | 0.01* | [1.08, 5.94] |
| Agree | 3.53 | 0.002** | [1.33, 5.73] |
| Strongly agree | 6.43 | <0.001*** | [3.26, 9.60] |
| Mental illness in friends & family | | | |
| No | Ref. | | |
| Yes | −2.36 | 0.004** | [−3.95, −0.77] |
| Exposure to psychotherapy | | | |
| No | Ref. | | |
| Yes | −2.12 | 0.03* | [−3.98, −0.26] |
| Contribution of the field | | | |
| Strongly agree | Ref | | |
| Agree | 1.98 | 0.04* | [0.05, 3.90] |
| Disagree/Strongly disagree | 2.69 | 0.02* | [0.52, 4.86] |

*Significance level $p < 0.05$.
**Significance level $p < 0.01$.
***Significance level $p < 0.001$.

this study aimed to examine the attitudes of adult and child psychiatrists, narcologists, mental health nurses and psychiatry residents toward individuals with mental health conditions. The overall stigma score among participants indicated a moderate level of stigma, though it was higher than in similar international studies. Older professionals, females and those in narcology had more stigma, as did those who viewed mental health work as less respectable. Lower stigma was linked to having friends or family with mental illness, prior psychotherapy exposure and a strong belief in the positive impact of mental health work.

Our study results are consistent with most of the previous research that has shown mental health professionals are not immune to stigmatizing beliefs (Schulze, 2007; Henderson et al., 2014). The overall OMS-HC scores indicate that stigma levels

among mental health workers in Kazakhstan are relatively high, with the sample's average total score being 40.1, 53.5% of the maximum. These findings are comparable to stigma scores reported in other countries. For instance, the mean stigma score among mental healthcare professionals in Bahrain was 36.8 (Al Saif et al., 2019); similar scores were observed in Belarus (35.01), Ukraine (35.02) and Latvia (35.98) (Őri et al., 2023a). In comparison, a multicenter study conducted across 32 European countries reported an average total score of 30.47 (Őri et al., 2023a), indicating that stigma levels in Kazakhstan are significantly higher.

The subscale scores of the OMS-HC were also significantly high. The score for Attitudes of healthcare providers toward individuals with mental illness was 14.02 (46.7% of maximum possible), the Disclosure/Help-seeking score was 11.25 (56% of maximum possible) and the Social Distance score was 14.9 (59.6% of maximum possible). Differing from this, the average total subscale scores for European countries were 11.11, 9.78 and 9.58, respectively (Őri et al., 2023a).

Notably, Kazakhstani mental health professionals scored particularly high on the following attitude statement: "More than half of people with mental illness don't try hard enough to get better" (mean = 3.27), reflecting a persistent stigma in this area. Our findings also emphasize that mental health specialists in Kazakhstan are particularly hesitant to disclose their personal mental health status to friends and colleagues, often perceiving themselves as weak if they cannot manage their challenges independently. The mean score for this subscale (11.25, 56% of the maximum possible) is consistent with previous research and comparable to findings from Belarus (11.31), Belgium (11.05), Latvia (11.00) and Ukraine (11.08) (Őri et al., 2023a). Furthermore, Kazakhstani professionals demonstrated a stronger inclination toward maintaining social distance from individuals with mental illness. The mean score for the Social Distance subscale (14.9, 59.6% of the maximum possible) was much higher than the average scores reported in Belarus (11.20), Latvia (11.71) and Ukraine (11.63) (Őri et al., 2023a). The underlying socio-cultural and systemic differences may affect stigmatizing attitudes of professionals across countries (Harangozo et al., 2014; Stefanovics et al., 2016).

Furthermore, the findings of our study revealed several significant associations, providing insights into the factors that can contribute to stigma in mental health settings. Age emerged as a significant predictor of stigma levels, with older professionals (≥55 years) demonstrating substantially higher stigma scores than their younger colleagues. This trend may reflect generational differences in attitudes toward mental health, as younger professionals are likely to have been exposed to more recent educational programs and awareness campaigns aimed at reducing stigma. Similar findings have been reported in European studies and Bahrain, where older specialists exhibited higher stigma scores (Al Saif et al., 2019; Őri et al., 2023a). On the contrary, our results contradict findings of the previous study from Turkey, which suggested that greater age and professional experience are associated with less stigmatizing attitudes among psychiatrists (Yavuz et al., 2020). This contrast highlights the complexity of stigma formation and suggests that factors beyond age, such as cultural influences and training approaches, may play a critical role in shaping professionals' perspectives on mental illness.

Gender differences in stigma levels were also observed, with male professionals reporting significantly lower stigma compared to their female counterparts. This finding contrasts with most of the existing literature, which suggests that male health providers hold significantly more stigmatizing views toward people with mental

illness, explained by theories of masculinity and stoicism, emphasizing emotional toughness and discouraging help-seeking (Almeida et al., 2022; Kaitz et al., 2022). Furthermore, previous studies showed that female health providers tend to report lower stigma scores, showing more empathy and willingness to socialize with the mentally ill people (Pascucci et al., 2017; Chiles et al, 2018). However, the evidence on gender differences in stigma is mixed (Kruse and Dodell-Feder, 2025), and many studies found no statistically significant differences between male and female participants (Destrebecq et al., 2018; Al Saif et al., 2019; Yavuz et al., 2020; Őri et al., 2023a). While studies from Western countries consistently show lower stigma in females, our findings align with a previous study from Tunisia, which showed that female providers exhibited higher scores on specific components of stigma, such as skepticism regarding treatment (Fekih-Romdhane et al., 2023). Findings of the current study may be explained by the specific cultural and healthcare context of Kazakhstan, where mental illness is highly stigmatized and the mental healthcare system remains focused on institutional care with a lack of community support services. While the majority of providers in the mental health field are represented by females, male professionals often occupy higher-ranking positions, granting them greater exposure to research-based, bio-psycho-social perspectives, which can contribute to lower stigma. In contrast, female professionals are more frequently involved in frontline patient care, where they experience greater exposure to severe and challenging cases, potentially reinforcing stigmatizing attitudes. Previous studies also reported that inpatient staff who primarily treat individuals with psychosis had the most negative attitudes (Hansson et al., 2013). Moreover, higher scores on the social distance subscale in our study may reflect heightened perceptions of risk due to greater contact with individuals who have chronic or recurrent problems, which may lead to a more realistic assessment of long-term outcomes and negative attitudes (Jorm et al., 1999), especially in situations involving close social proximity, such as living as a neighbor or contact with children.

The professional field had a clear impact on stigma levels, with specialists in narcology showing significantly higher stigma compared to those in psychiatry and psychotherapy. In many European countries, psychiatry and addiction treatment are fully integrated disciplines. However, in post-Soviet nations such as Kazakhstan, despite the formal combination of these specialties during residency training, they continue to function as separate fields in practice, making direct comparisons difficult. Research has consistently shown that individuals with substance use disorders face some of the highest levels of stigma (Schomerus et al., 2011) even within healthcare settings (van Boekel et al., 2013; Valdesalici et al., 2024). The attribution theory suggests that the way people interpret the cause of a stigmatized identity influences their reactions toward the stigmatized individual (Weiner, 1995). Substance use disorders are often perceived as a controllable condition, compared to other uncontrollable conditions such as genetic disorders or severe mental illness, and therefore the response may be associated with higher stigma levels, including blame, anger or social exclusion, even from the side of mental health professionals. To reduce the stigma and barriers to effective treatment, educational interventions are needed to improve the understanding of drug dependence as not only a social, but rather as a health problem which should be treated and evaluated like any other complex chronic disease influenced by a combination of genetic, developmental and environmental factors (McLellan et al., 2000).

Perceptions of a profession in mental health played a crucial role in determining stigma levels. Professionals who considered the field less respectable demonstrated significantly higher stigma scores. Additionally, professionals who doubted the positive contribution of their field demonstrated significantly higher stigma scores compared to those who strongly affirmed its value. Research has demonstrated that psychiatrists face greater stigmatizing attitudes and discriminatory behaviors compared to other medical specialists (Gaebel et al., 2015). Addressing occupational stigma among psychiatrists is essential, as their perceptions can directly influence the quality of care they provide and affect individuals' willingness to seek mental health support (Shi et al., 2023). Furthermore, occupational stigma is closely linked to higher burnout rates and lower patient satisfaction (Verhaeghe and Bracke, 2012). Despite these connections, the relationship between occupational stigma and psychiatrists' attitudes toward patients remains largely unexplored, highlighting an important area for future research.

In our study, burnout did not emerge as a predictive factor for stigmatizing attitudes. However, a substantial body of research consistently demonstrates a strong association between stigma and burnout among mental health professionals (Gibb et al., 2010; Henderson et al., 2014; Solmi et al., 2020; Yavuz et al., 2020), with some authors suggesting that stigmatizing attitudes may constitute a component of the burnout process itself (Holmqvist and Jeanneau, 2006). Across specific burnout dimensions, emotional exhaustion appears to play a central role, showing a strong association with negative attitudes and the highest predictive value for stigma (Gibb et al., 2010; Yavuz et al., 2020). In parallel, low personal accomplishment is consistently linked to higher levels of stigma (Gibb et al., 2010; Solmi et al., 2020). Conversely, higher personal accomplishment is associated with more accepting attitudes toward patients (Holmqvist and Jeanneau, 2006). Overall, these findings highlight the complexity of the relationship between burnout and stigma and underscore the need for further targeted research to clarify the underlying mechanisms.

Personal exposure to mental illness was linked to lower stigma levels among mental health professionals in our study. Those with close friends or family members affected by mental illness displayed significantly lower stigma scores. A study in Brazil similarly found that infrequent contact with a family member diagnosed with schizophrenia was associated with less stigma (Loch et al., 2013a). These findings align with broader research indicating that personal experience, whether through one's own mental health challenges or close relationships with those affected, plays a crucial role in reducing mental health-related stigma (Őri et al., 2022, 2023a). This supports the idea that fostering both personal and professional engagement with individuals experiencing mental illness can lead to more compassion and decreased stigma (Alexander and Link, 2003; Henderson et al., 2016).

Specialists who also engage in psychotherapy demonstrated lower levels of stigma. These findings correlate with European studies, which indicate that openness to professional discussions, such as participation in case discussion groups, supervision or Balint groups, is strongly associated with more positive attitudes toward individuals with mental illness (Őri et al., 2022, 2023a). Likewise, research from Turkey emphasizes the significance of psychotherapy training in reducing stigma (Yavuz et al., 2020), suggesting that deeper involvement in therapeutic approaches promotes greater empathy and understanding.

## *Implications for practice*

These findings carry significant implications for mental health education, policy and professional development. It is crucial to

address both the occupational stigma directed at mental health professionals and the perceived stigma experienced by individuals working in the field. Strengthening the professional identity of mental health workers and enhancing the prestige of the field by emphasizing its societal contributions could play a primary role in reducing stigma. Efforts to promote recognition and respect for mental health professionals may not only improve their well-being but also contribute to more compassionate and effective patient care. In addition, stigma reduction initiatives should prioritize professionals with limited personal exposure to mental illness, incorporating programs that facilitate direct engagement with individuals who have lived experiences. Such interactions can foster empathy and challenge misconceptions.

### Strengths and limitations

One of the key strengths of this study is that it is the first to investigate stigma levels toward people with mental disorders within the healthcare workforce in Kazakhstan, contributing novel insights to a previously unexplored context. The study was conducted with a sufficiently large sample size ($n$ = 348), ensuring adequate statistical power. Another major strength is the use of the validated OMS-HC to assess stigma, which enhances the reliability and comparability of findings. Additionally, the study examined a broad range of potential determinants of stigma, allowing for a more comprehensive understanding of the factors influencing attitudes within the healthcare setting.

Despite these strengths, the study has several limitations. The cross-sectional design restricts the ability to establish causal relationships between determinants and levels of stigma. Furthermore, the use of self-reported questionnaires introduces the potential for social desirability bias, whereby participants may underreport stigmatizing attitudes or overreport socially acceptable responses. This could lead to an underestimation of the true prevalence of stigma among healthcare professionals. Although the translated version of the OMS-HC demonstrated good reliability, the lack of assessment of construct validity represents a limitation, and future studies are recommended to address this aspect. Additionally, the use of non-probability (snowball and convenience) sampling methods may have introduced selection bias, potentially attracting participants with greater interest in mental health topics or more favorable attitudes. As a consequence, the applicability of these outcomes to all healthcare professionals in Kazakhstan may be constrained, particularly for those working in more remote or resource-limited settings.

### Future research directions

Further research is needed to explore stigma levels across different mental health conditions, as attitudes toward disorders such as schizophrenia, substance use disorders and other psychiatric diagnoses may vary significantly. Notably, this study found particularly high levels of stigma among narcologists, highlighting the need for focused investigation into stigma toward people with addictions. Substance use disorders often carry additional moral and societal judgments, which may exacerbate negative attitudes even among mental health professionals. Understanding the specific drivers of this elevated stigma within addiction treatment fields is crucial for developing effective, condition-specific anti-stigma interventions.

In addition, future studies should examine stigma across diverse populations, including the general public, general practitioners and other healthcare providers, to better inform targeted and population-specific strategies. Longitudinal research designs would allow for the assessment of how stigma evolves over time, particularly in response to educational or organizational interventions. Moreover, qualitative research could provide rich insights into the contextual and cultural factors that influence stigma, as well as uncover the reasons for variations in stigma across professional disciplines. Such multi-method approaches are essential to inform the development of comprehensive and sustainable stigma-reduction strategies within Kazakhstan's healthcare system and beyond.

### Conclusion

This study highlights the persistence of stigma among mental health professionals in Kazakhstan, with overall stigma levels higher than those reported in other previous studies. Key findings indicate that older professionals, those working in narcology and individuals who perceive the profession as less respectable and impactful, possess higher stigma. Gender differences were also observed, with male professionals reporting lower stigma. Additionally, those with personal experiences of mental illness and engagement in psychotherapy demonstrated lower stigma levels.

These findings emphasize the critical need for directed efforts to combat stigma, particularly for professionals working with addictions. Addressing occupational stigma, promoting psychotherapy training and strengthening the professional perception of mental health workers are essential steps toward improving attitudes toward patients with mental illness and patient care. Further research is needed to explore stigma variations across mental health conditions and different healthcare populations to develop future interventions.

**Open peer review.** To view the open peer review materials for this article, please visit http://doi.org/10.1017/gmh.2026.10211.

**Supplementary material.** The supplementary material for this article can be found at http://doi.org/10.1017/gmh.2026.10211.

**Data availability statement.** The data that support the findings of this study are available on request from the corresponding author. The data are not publicly available due to privacy or ethical restrictions.

**Acknowledgments.** The authors thank all respondents for their interest and voluntary participation in this study.

**Author contribution.** AA: Conceived and conducted the study, managed data collection, performed statistical analyses and wrote the original draft of the manuscript. YN: Conceived the study, provided methodological expertise and critically reviewed the manuscript. NY: Managed data collection and quality control, critically reviewed the manuscript. RA: Conceived the design and methodology of the study, supervised the project, revised and edited the manuscript. All authors have read and approved the final manuscript for submission.

**Financial support.** This research project received no financial support.

**Competing interests.** All authors declare no conflicts of interest.

**Ethics statements.** All procedures were conducted under the institutional guidelines and ethical standards. The Research Ethics Committee at the Nazarbayev University School of Medicine has reviewed and approved this study (Re:2024Nov#03).

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
