## [Reviewer Report]

Dear Authors, the article is great, written with attention to detail and adherence to the STROBE reporting guideline (I think you have to mention it in the Method section). The topic itself is extremely important, also taking into account the lack of data and information about mental health in Central Asian countries. It is worth mentioning that the instrument used for the stigma assessment was thoroughly adapted with permission from the developers.

---

## [Reviewer Report]

It is my pleasure to review this manuscript and to read a study examining the attitudes of mental health professionals in Kazakhstan toward people with mental illness. Overall, the paper is well-written and uses appropriate terminology. The topic is significant, especially as no previous studies on this subject have been conducted in Kazakhstan. This makes the work highly relevant and interesting to the readership, and importantly, valuable for the country itself. There is a clear gap in the literature regarding mental health stigma among healthcare workers, and this study addresses it effectively. As someone working in and interested in the field of stigma research, I anticipate using the results of this study and citing it in my future work.

Abstract:

The use of the word “prevalence” may be misleading. While the total number of healthcare providers in the country is provided, the sample is not necessarily representative due to the sampling methodology and lack of coverage across all regions, age groups etc. Additionally, without clear categorical definitions of who is or is not stigmatizing, prevalence cannot be accurately calculated. I recommend removing the term “prevalence” from the abstract.

Impact Statement:

The statement that “societal misunderstandings” contribute to stigma is imprecise. I suggest expanding this to explicitly acknowledge the roles of prejudice, negative stereotypes, and discriminatory behavior in this context.

Methods:

As three languages were used in the survey, it is important to report how many responses were completed in English, Kazakh, and Russian. While ethnicity percentages are provided, the language breakdown should also be included.

It is also important to state whether the OMS-HC scale has been validated in all three languages. While the Russian version is indicated as validated, please provide a citation for it. For the Kazakh version, please specify whether it has been validated. If yes, cite it, if it has not, this should be mentioned in both the Methods section and the Limitations section.

The statistical analyses are appropriate.

Results and discussion:

• In Table 1, you note that “total frequencies may vary because of missing values,” yet in the Methods, it is stated that responses with missing values were deleted. Please clarify this discrepancy.

• It is interesting that burnout was not related to stigma scores. This is a negative but very noteworthy finding. While Solmi et al. are mentioned in the Discussion, this result is not discussed. I recommend highlighting and interpreting this finding in the Discussion section.

• Regarding gender differences, it is somewhat surprising that women show higher stigma scores; however, such findings have been reported in certain contexts. Thank you for addressing potential cultural explanations, as this provides valuable insight for readers. I would encourage you to further explore the literature and include studies reporting similar patterns, as evidence on gender differences in stigma is mixed, with some studies showing higher stigma among men, while others among women. This would help to better contextualize your findings within the existing body of research.

• It is excellent that the OMS-HC was translated into Kazakh. Consider including the Kazakh translation in the supplementary materials.

These are mostly minor issues. The manuscript is well-written and makes a valuable contribution to the literature, especially given the scarcity of research from Kazakhstan, a large and under-studied country in this field. I am grateful to see these findings added to the global literature.

---

## [Editor Report]

Thank you for submitting your manuscript titled “Stigmatizing attitudes toward mental illness among mental health professionals in Kazakhstan: a cross-sectional study” to Cambridge Prisms: Global Mental Health. The reviewers agree that your study addresses an important topic; however, they have suggested several minor revisions to improve the depth and clarity of the paper. Please address these comments and submit your revised manuscript for final consideration.

---

## [Editor Report]

Thank you for submitting your revision of the manuscript titled: “Stigmatizing attitudes toward mental illness among mental health professionals in Kazakhstan: a cross-sectional study”. The revisions successfully address all points raised during the review process, and the manuscript is now suitable for publication in its current form.